**Data Availability Statement:** All relevant data are within the paper and its Supporting information files.

# Characterization of regional variation of bone mineral density in the geriatric human cervical spine by quantitative computed tomography

**Ryan S. Garay[1]**, **Giovanni F. Solitro[2]**, **Kenrick C. Lam[3]**, **Randal P. Morris[3]**, **Abeer Albarghouthi[4]**, **Ronald W. Lindsey**[3], **Loren L. Latta[4,5]**, **Francesco Travascio**[1,4,5,6] *

**1** Department of Mechanical and Aerospace Engineering, University of Miami, Coral Gables, Florida, United States of America, **2** Louisiana State University Health-Shreveport, Shreveport, Louisiana, United States of America, **3** University of Texas Medical Branch, Galveston, Texas, United States of America, **4** Max Biedermann Institute for Biomechanics, Mount Sinai Medical Center, Miami Beach, Florida, United States of America, **5** Department of Orthopaedic Surgery, University of Miami, Miami, Florida, United States of America, **6** Department of Industrial Engineering, University of Miami, Coral Gables, Florida, United States of America

* f.travascio@miami.edu

## Abstract

### Background

Odontoid process fractures are among the most common in elderly cervical spines. Their treatment often requires fixation, which may include use of implants anteriorly or posteriorly. Bone density can significantly affect the outcomes of these procedures. Currently, little is known about bone mineral density (BMD) distributions within cervical spine in elderly. This study documented BMD distribution across various anatomical regions of elderly cervical vertebrae.

### Methods and findings

Twenty-three human cadaveric C1-C5 spine segments (14 males and 9 female, 74±9.3 y. o.) were imaged via quantitative CT-scan. Using an established experimental protocol, the three-dimensional shapes of the vertebrae were reconstructed from CT images and partitioned in bone regions (4 regions for C1, 14 regions for C2 and 12 regions for C3-5). The BMD was calculated from the Hounsfield units via calibration phantom. For each vertebral level, effects of gender and anatomical bone region on BMD distribution were investigated via pertinent statistical tools.

Data trends suggested that BMD was higher in female vertebrae when compared to male ones. In C1, the highest BMD was found in the posterior portion of the bone. In C2, BMD at the dens was the highest, followed by lamina and spinous process, and the posterior aspect of the vertebral body. In C3-5, lateral masses, lamina, and spinous processes were characterized by the largest values of BMD, followed by the posterior vertebral body.

**Funding:** The author(s) received no specific funding for this work.

**Competing interests:** The authors have declared that no competing interests exist.

## Conclusions

The higher BMD values characterizing the posterior aspects of vertebrae suggest that, in the elderly, posterior surgical approaches may offer a better fixation quality.

## Introduction

Odontoid process fractures are one of the most common cervical spine fractures in the elderly and are associated with increased morbidity and mortality [1–8]. Recent evidence suggests that there is a survival advantage and a trend toward improved quality of life after operative intervention as compared to non-operative treatment in geriatric patients with odontoid fractures [9,10]. Options for operative management of odontoid process fractures consist of reduction and internal fixation with an anterior odontoid screw or posterior atlantoaxial arthrodesis. Anterior odontoid screw fixation, although motion preserving, is associated with a high complication rate in the elderly due to bone fragility and cervical spine stiffness [11]. Posterior atlantoaxial arthrodesis is suitable for most fracture patterns but at the cost of range of motion [12]. Bilateral C1-C2 screw or screw-rod constructs have become very popular posterior atlantoaxial fixation techniques in recent years [13,14]. Although the clinical outcomes of these posterior fixation techniques among all odontoid fracture patients have been examined [15,16], the quality of fixation achieved by each modality in elderly osteoporotic spines has not been established. Moreover, little is currently known about bone mineral density (BMD) distribution variations within the cervical spine in the elderly and how those variations may affect different fixation techniques.

The objective of this study was to document the BMD distribution across various anatomical regions of the elderly cervical vertebrae at different levels. The rationale for pursuing such characterization is based on the premise that cervical spine bone quality distribution data will assist in determining the optimal cervical spine fixation technique(s) in geriatric patients. This has been accomplished by measuring region-specific BMD via quantitative CT analysis of human cadaveric specimens. A similar approach has been successfully used in other studies aimed at characterizing bone mineral distribution across lumbar vertebral bodies [17], and adult and young adult cervical spines [18,19].

## Materials and methods

### Specimens

Twenty-three intact fresh human cadaver specimens (14 males and 9 female, age 74 ± 9.3 y.o., BMI 21.6 ± 5.5), including the cervical spine segment (C1-C5), were obtained from a tissue bank (United Tissue Network, Inc., St. Petersburg, FL). As donors were not identified, this study was IRB exempted as per the National Institute of Health (NIH) guidelines (exemption 4). The specimens were wrapped in saline soaked towels, hermetically sealed and stored at -20°C prior to imaging. Sealed specimens were thawed in air overnight prior to scanning.

### QCT image acquisition

Specimens were imaged via single-energy QCT using a clinical computed tomography (CT) scanner (LightSpeed VCT, GE Medical Systems, Chicago, IL). The QCT image volume included the entire head of the specimen, though only QCT images of the cervical segment C1-C5 were used in this analysis. CT scanning parameters included: bone standard

reconstruction algorithm, axial scanning plane, 120 kV tube voltage, 99.0 mA tube current, 0.8 second scan time, ~200 slices, 1.25 mm slice thickness, 0.5 x 0.5 mm$^2$ in-plane pixel resolution. To mimic the physiologic setting, heads were orientated in a supine position when imaged. A QCT scan of a solid dipotassium phosphate (K2HPO4) phantom (QCT Pro; Mindways Software Inc, Austin, TX, USA) was used to convert grayscale CT Hounsfield units (HU) to an equivalent volumetric bone mineral density (vBMD). The CT HU values were converted to vBMD using a previously validated technique [20]. The mean HU values within each of the reference phantom cylinders were calculated. Subsequently, a regression equation ($R^2 > 0.99$), derived from the mean HU values and known reference cylinder densities, was used to convert HU to equivalent volumetric BMD.

## Segmentation and region identification

A flowchart of the sequence of operations for achieving the segmentation and partition of the vertebrae in distinct bone regions is reported in Fig 1. Briefly, for each cervical spine, vertebrae were individually segmented (3D Slicer v.4.8.1) [21]. In a similar fashion of a previously reported technique [17], the segmented vertebrae were exported as STL files into a dedicated software for mesh smoothening (Autodesk Meshmixer v3.5) and, subsequently, into a CAD software (Autodesk Fusion 360 v.2.0.9313) to be partitioned in regions according to predefined anatomical landmarks and cervical level. More specifically, for C1, a total of 4 regions were created by splitting the vertebra along a medial line from the anterior to the posterior tubercle; 2 regions included the anterior arches, the articular surfaces, and the transverse processes, and the other 2 regions comprised the posterior arches. For C2, 14 regions were created: the vertebral body was split in 8 octants; 2 more regions included the lateral masses and transverse processes on their respective sides; 2 regions comprised the lamina through to the spinous process; and 2 other regions represented the superior and inferior portions of the dens. For C3-C5, a total of 12 regions were created: 8 regions for the vertebral body; 2 more regions were added for the lateral masses and transverse processes on their respective sides; and 2 regions included the lamina through to the spinous process. A graphical representation of the bone regions for each vertebral level is reported in Fig 2. Subsequently, STL models of the partitioned bone regions were imported back into 3D Slicer, and their BMD values were determined using the 'Segment Statistics' module.

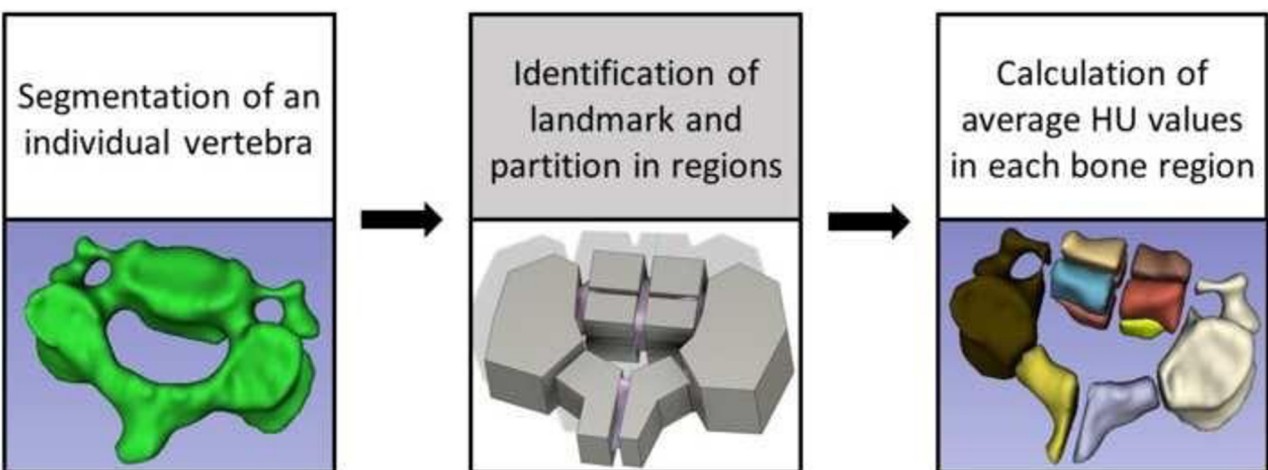

**Fig 1. Workflow to segment and partition vertebrae in bone regions.**

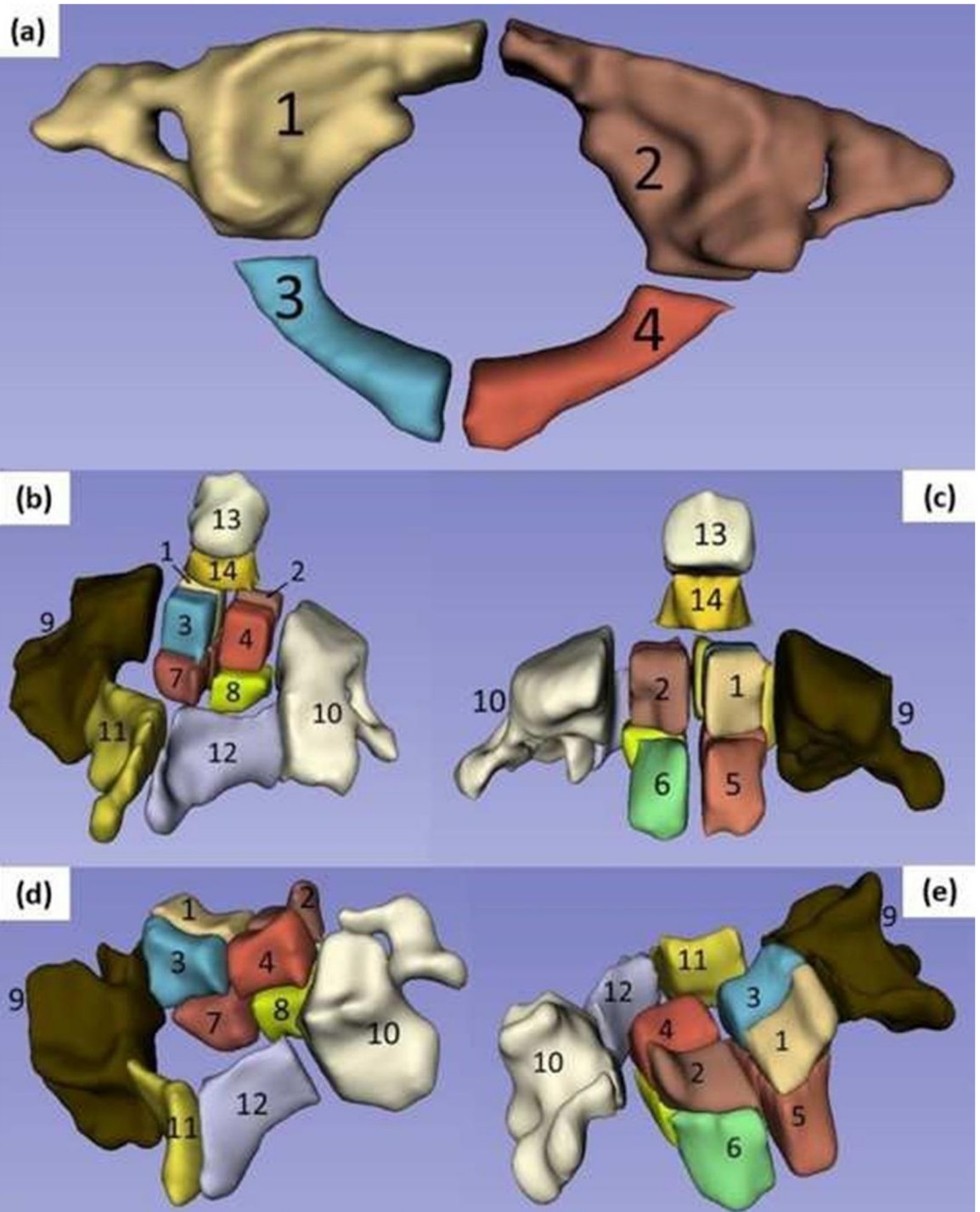

**Fig 2. Identification of bone regions for each vertebral level: (a) bone regions of C1; (b) bone regions of C2, posterior view; (c) bone regions of C2, anterior view; (d) bone regions of C3-5, posterior view; (e) bone regions of C3-5, anterior view.**

## Statistical analysis

A preliminary inspection of the data via Anderson-Darling test indicated that all the data collected were normally distributed. Accordingly, all the data were reported in terms of mean ± standard deviation or, when appropriate, in terms of 95% confidence interval. The morphology of C1 and C2 vertebrae present major differences with respect to the C3-C5 levels. Therefore, data pertinent to C1 and C2 were analyzed separately, while those of C3-C5 were combined. For each vertebral level, 2-sample mean t-tests were conducted to investigate gender dependent differences in BMD or bone volume. One-way ANOVA followed by post-hoc Tukey test was used to investigate any significant effect of vertebral level (5 levels) on BMD or bone volume. When investigating regional distribution of BMD within vertebral levels, female and male data were initially analyzed separately, and then combined if no significant gender-dependent difference was observed. Specifically, when investigating the mineral density of C1, one-way ANOVA followed by post-hoc Tukey test was used to individuate any significant effect of bone region (4 regions) on the BMD distribution within the vertebra. The same approach was used for investigating the effect of bone region (14 regions) on the BMD of C2. When analyzing data from C3, C4 and C5, BMD data were initially separated by gender and vertebral level, and then combined if no significant difference due to gender or vertebral level was observed. One-way ANOVA followed by post-hoc Tukey test was used to individuate any significant effect of bone region (12 regions) on the BMD distribution. For all tests performed, the level of significance was set to $\alpha = 0.05$. Outliers elimination, if needed, was conducted via Grubbs test.

## Results

For all the vertebral levels investigated, BMD values of female vertebrae were larger than those found in male samples, although differences were not statistically significant (p-value $> 0.05$). For both female and male vertebrae, the largest values of BMD were found in C1 (CI: 435.7, 587.3 mg/cm$^3$), followed by C4 (CI: 397.6, 549.3.3 mg/cm$^3$) and C2 (CI: 381.6, 533.2 mg/cm$^3$). The lowest values corresponded to C3 (CI: 353.3, 504.9 mg/cm$^3$) and C5 (CI: 337.8, 492.8 mg/cm$^3$) levels, see Fig 3. For each vertebral level, the mean values of bone volumes of male vertebrae were larger than those of female ones, but not statistically different (p-value $> 0.05$). For both female and male samples, the mean volumes of C1 (CI: 12.37, 14.8 cm$^3$) and C2 (CI: 13.99, 16.42 cm$^3$) were significantly larger (p-value $< 0.001$) than those of C3 (CI: 9.41, 11.84 cm$^3$), C4 (CI: 9.58, 12.0 cm$^3$) and C5 (CI: 9.37, 11.85 cm$^3$), see Fig 4.

When investigating the regional distribution of bone density in C1, it was found that the largest values of BMD were observed in the posterior portion of the vertebra (regions 3 and 4) for both female and male samples. Statistically significant differences were found in only male samples when comparing BMD values in regions 1 and 2 to those of regions 3 and 4 (p-value $< 0.001$), see Fig 5. For each region investigated, BMD of female samples was not significantly different from that of male samples (p-value $> 0.05$). When combining female and male samples together, BMD values in regions 3 and 4 (693.7 and 720.3 mg/cm$^3$, respectively) was approximately double the BMD observed in regions 1 and 2. A summary of the descriptive statistics for each bone region, together with statistical grouping is reported in Table 1.

Bone densities in C2 female and male samples were similar, see Fig 6. When combining female and male data together, significant regional variations of BMD were observed, with the highest values found in the dens (608.4 and 823 mg/cm$^3$), followed by the lamina and spinous process (386.1 and 393.8 mg/cm$^3$) and the posterior aspect of the vertebral body (366.3 and 372.5 mg/cm$^3$). No statistically significant differences were found in the remaining regions (p-value $> 0.05$). Descriptive statistics and grouping are reported in Table 2.

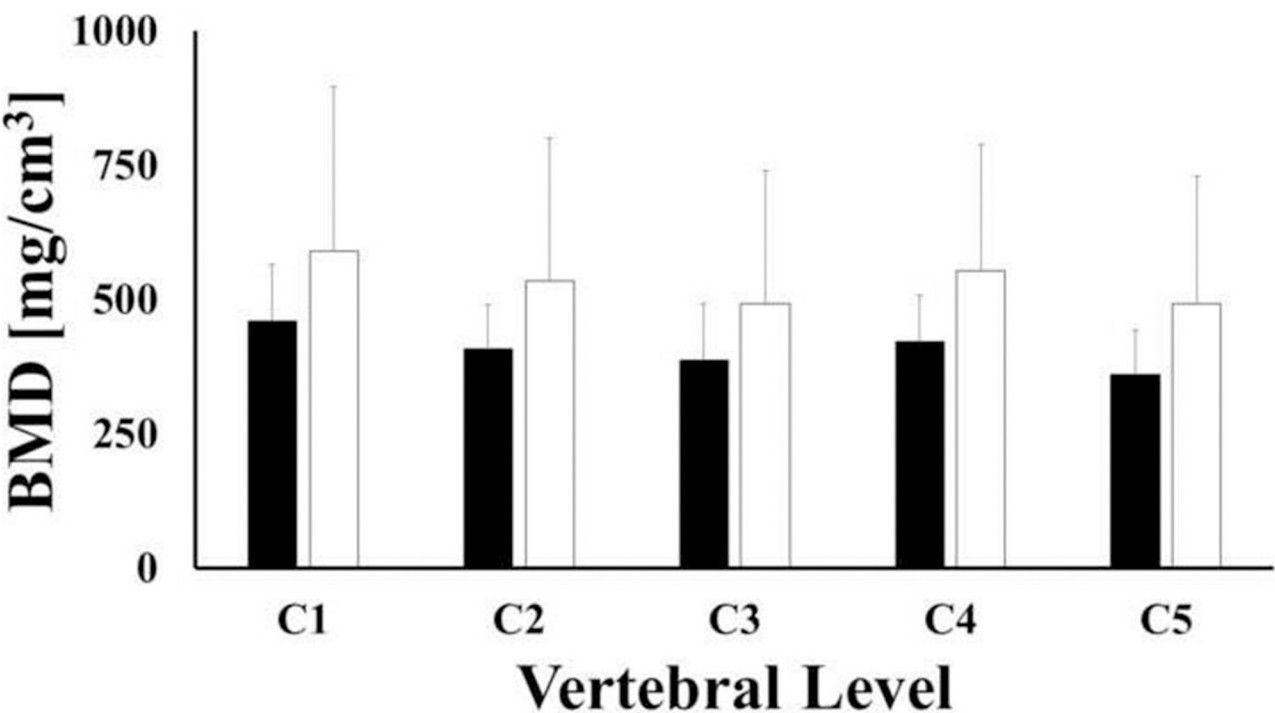

**Fig 3. Mean values of bone mineral density across vertebral levels for female (white) and male (black) samples.** Bars indicate one standard deviation.

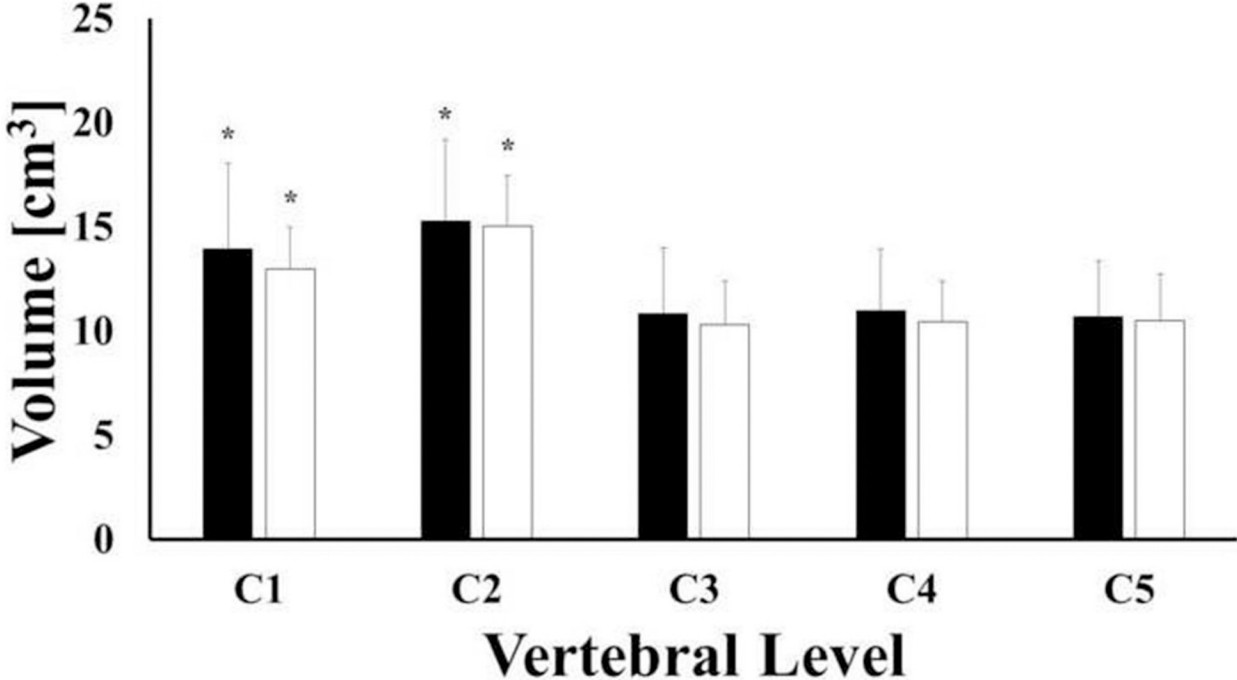

**Fig 4. Mean values of bone volume across vertebral levels for female (white) and male (black) samples.** Statistical significance (p-value < 0.05) is denoted by (*). Bars indicate one standard deviation.

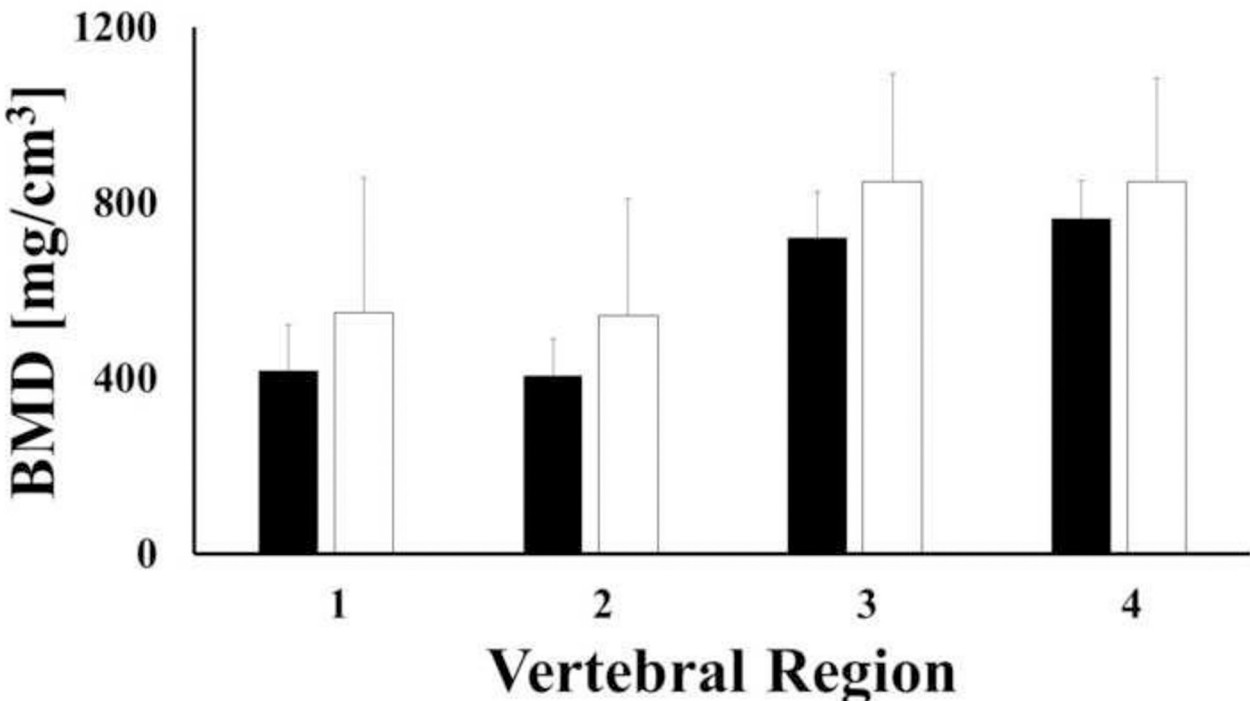

**Fig 5. Mean values of bone mineral density across regions of C1 for female (white) and male (black) samples.** Bars indicate one standard deviation.

The mean values of BMD in bone regions of C3, C4 and C5 are reported in Fig 7. No statistical differences were observed among female and male samples. The lateral masses, together with the lamina and the spinous processes (from 470.3 to 517.6 mg/cm$^3$) were characterized by the largest values of BMD, followed by the posterior portion of the vertebral body (from 380.5 to 400.1 mg/cm$^3$). The lowest BMD values were found in the anterior portion of the vertebral body (from, 278.2 to 318.4 mg/cm$^3$). A summary of the descriptive statistics for each bone region, together with statistical grouping is reported in Table 3.

## Discussion

Although not always statistically significant, the results reported in this study suggest that gender may influence both mineral density and volume of cervical vertebrae. Specifically, the authors found out that the average values of vertebrae BMD in female specimens were larger than those found in male specimens across all levels. The opposite gender trend was observed

**Table 1. Descriptive statistics of BMD in C1.** Values of BMD are reported in terms of mg/cm$^3$.

| Region | N | Mean | SD | Min | Max | Group |
|---|---|---|---|---|---|---|
| 1 | 23 | 357.3 | 138.3 | 162.9 | 692.7 | A |
| 2 | 23 | 354.8 | 143.8 | 176.3 | 742.3 | A |
| 3 | 23 | 693.7 | 316 | 214.5 | 1391.3 | B |
| 4 | 23 | 720.3 | 334.2 | 135.3 | 1352.7 | B |

Letters identify statistical groups.

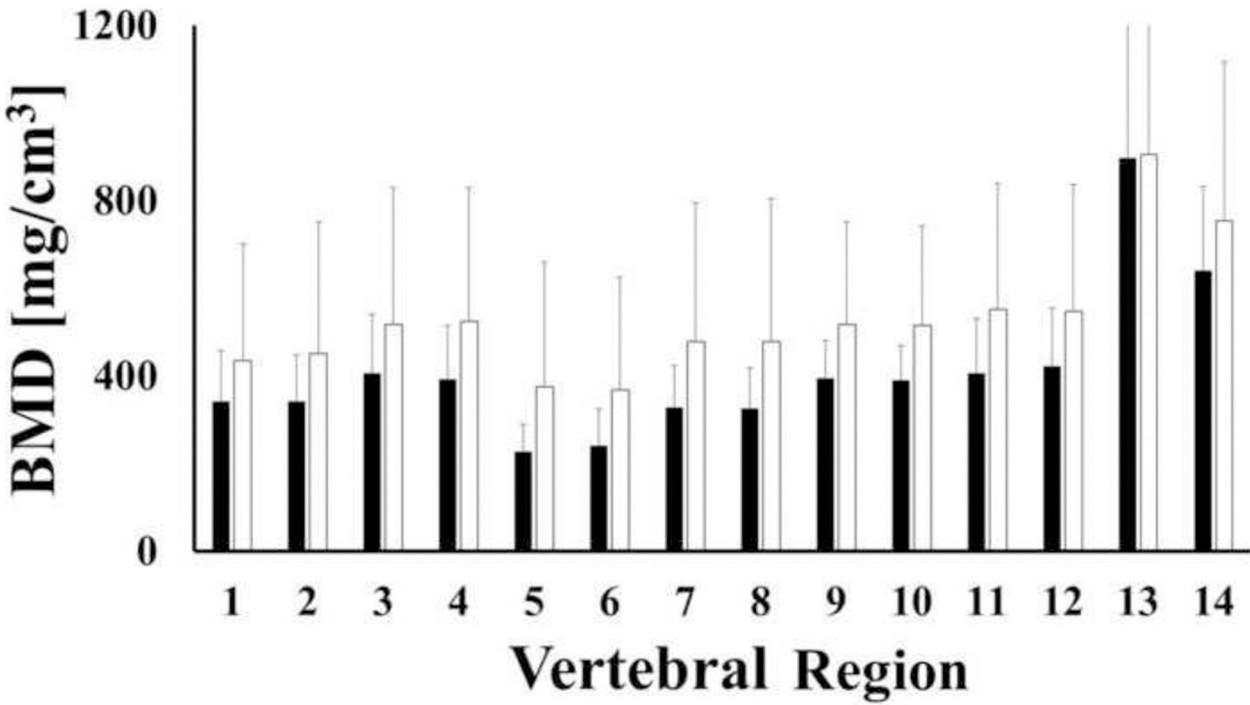

**Fig 6. Mean values of bone mineral density across regions of C2 for female (white) and male (black) samples.** Bars indicate one standard deviation.

for the average vertebral volumes, see Figs 3 and 4. Despite the fact that these results are in agreement with several similar prior studies [18,19,22–25], they contradict the conventional wisdom that associates lower BMD to females compared to males. As speculated by Anderst and co-workers [18,19], in contrast to gender biomechanical load variations inherent in other

**Table 2. Descriptive statistics of BMD in C2.** Values of BMD are reported in terms of mg/cm$^3$.

| Region | N | Mean | SD | Min | Max | Group |
|---|---|---|---|---|---|---|
| 1 | 23 | 301.4 | 189.4 | 73.4 | 860.7 | AB |
| 2 | 23 | 307.5 | 207.2 | 55.2 | 874.8 | AB |
| 3 | 23 | 372.5 | 222.5 | 98.9 | 1040.8 | A |
| 4 | 23 | 366.3 | 216.8 | 103.1 | 1022.7 | A |
| 5 | 23 | 140.7 | 61.0 | 37.2 | 257.2 | B |
| 6 | 23 | 213.6 | 179.3 | 40.8 | 669.9 | AB |
| 7 | 23 | 310.8 | 216.8 | 77.7 | 971.9 | AB |
| 8 | 23 | 249.2 | 111.6 | 78.0 | 496.6 | AB |
| 9 | 23 | 340.5 | 120.0 | 159.6 | 606.0 | AB |
| 10 | 23 | 338.7 | 121.7 | 158.5 | 677.0 | AB |
| 11 | 23 | 386.1 | 210.7 | 87.3 | 980.4 | A |
| 12 | 23 | 393.8 | 210.7 | 96.8 | 1056.3 | A |
| 13 | 23 | 823.0 | 329.4 | 314.3 | 1470.9 | C |
| 14 | 23 | 608.4 | 269.8 | 202.4 | 764.6 | D |

Letters identify statistical groups.

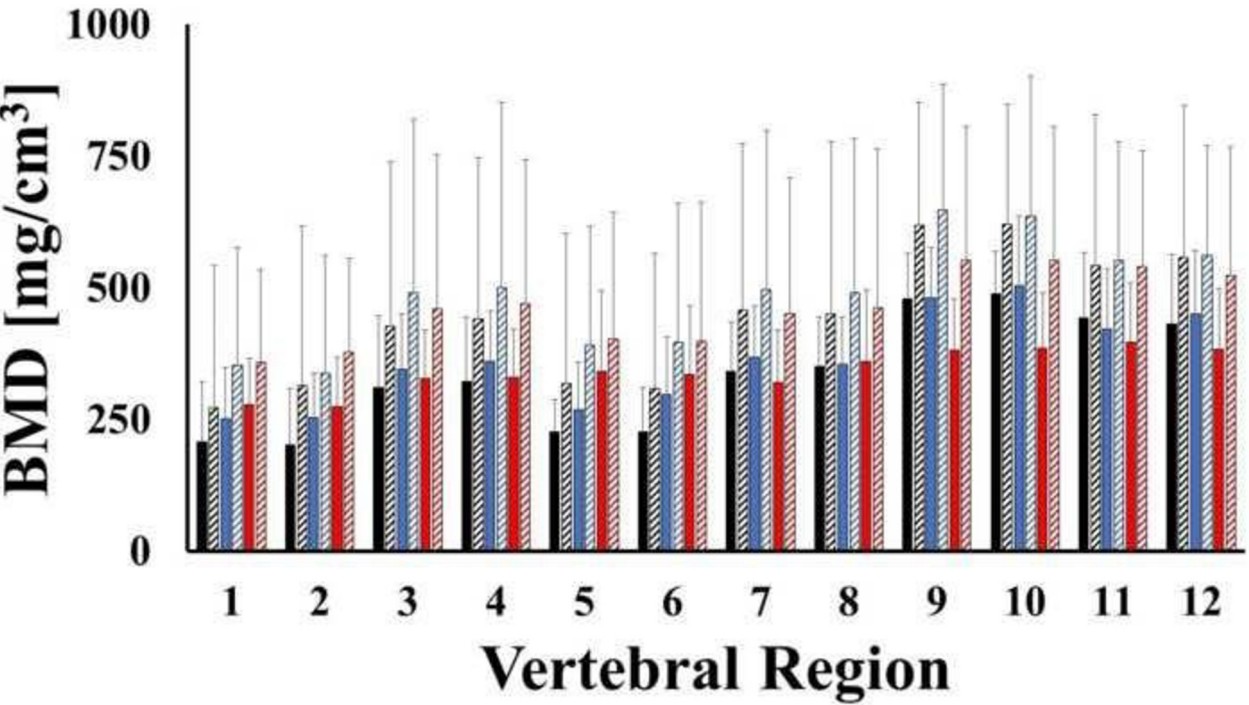

**Fig 7. Mean values of bone mineral density across regions of C3 (black), C4 (blues) and C5 (red) for male (solid) and female (stripes) samples.** Bars indicate one standard deviation.

musculoskeletal regions (e.g. hip, lumbar spine, etc.), the mechanical loads on female and male cervical spines might be similar. Our data also indicate that the female cervical vertebrae are smaller than male vertebrae (Fig 4). Accordingly, the magnitude of the mechanical stress acting on female cervical vertebrae is greater and requires higher bone strength, which is positively related to bone density [26,27]. This could possibly explain why our average female specimen BMD values were higher than those from males.

**Table 3. Descriptive statistics of BMD in C3-5.** Values of BMD are reported in terms of mg/cm$^3$.

| Region | n | Mean | SD | Min | Max | Group |
|---|---|---|---|---|---|---|
| 1 | 69 | 278.3 | 139.5 | 81.7 | 838.9 | D |
| 2 | 69 | 282.8 | 150.9 | 69.4 | 839.7 | D |
| 3 | 69 | 380.5 | 198.5 | 162.5 | 1195.6 | BCD |
| 4 | 69 | 390.3 | 206.3 | 159.4 | 1248.1 | BC |
| 5 | 69 | 315 | 171.9 | 74.5 | 837.2 | CD |
| 6 | 69 | 318.4 | 176 | 76.2 | 975.2 | CD |
| 7 | 69 | 393.6 | 193.6 | 125.6 | 1143.3 | BC |
| 8 | 69 | 400.1 | 197 | 152.4 | 1102.4 | BC |
| 9 | 69 | 511.3 | 192.4 | 182.6 | 1081.3 | A |
| 10 | 69 | 517.6 | 194.4 | 245.8 | 1142 | A |
| 11 | 69 | 470.3 | 179.1 | 211.1 | 928 | AB |
| 12 | 69 | 472.5 | 180.5 | 213.3 | 908.8 | AB |

Letters identify statistical groups.

The average volumes of the vertebrae measured in this study were comparable in magnitude and trend to those previously reported, with C2 being the largest (~16 cm$^3$), followed by C1 (~14cm$^3$), and C3-C5 (~10 cm$^3$) [18,19]. In contrast, for all the vertebral levels investigated, the BMD values were smaller than those previously reported in similar analyses [19,28]. However, it should be considered that the demographic composition of the vertebral samples used in those studies included young adults and/or adults. The principal contribution of this study is that it specifically targets geriatric human cervical spine (age 74 ± 9.3 y.o.). Therefore, lower BMD levels, compared to a younger specimen population, would be expected. Also, in agreement with previous studies [18], our results demonstrate that the highest BMD values were found in C1. Furthermore, the second highest BMD values were found in C4 and subsequently decreased at relative rostral and caudal vertebral levels. Previous studies have reported a different trend, with the largest BMD values detected at C5 compared to C2-C4 and C6-C7 segments [22,23,28–30]. These differences in BMD by vertebral levels may be due to variations is an individual's physiological conditions, which would alter the magnitude of mechanical load experienced by a particular vertebra over time. The higher value of BMD in C5 reported in young and adults can be attributed to this level being exposed to a larger mechanical load [18,19], as Wolff's Law would predict. The fact that the geriatric vertebrae used in this study demonstrated higher BMD values at C4 compared to C5 may suggest a different in-vivo mechanical load distribution across vertebral levels in the elderly spine. This would be reasonable to expect in view of the postural changes that typically occur in the neck with ageing [31].

The distributions of BMD across different anatomical regions of each vertebral level were also investigated. The choice of the specific anatomical region subdivision was motivated to document BMD quality in the anterior and posterior vertebral body (8 regions), as well as the lateral masses (2 regions) since these are the locations where fixation hardware is usually implanted. Two additional regions (including the lamina and the spinous process) were also investigated as this is where bone grafts can be harvested from. In general, for each level considered, the highest BMD values were measured in the posterior regions of the bone, while the lowest BMD values were detected in the anterior regions. Specifically, in C3-C5, the average BMD of the lateral masses, the lamina and the posterior vertebral body were 65%, 55% and 30% larger than those found in the anterior vertebral body, respectively (Fig 7 and Table 3). Differences of these magnitudes across similar anatomical regions have been reported in studies on both young and adult cohorts [18,19]. The trends of bone density distribution in C2 were similar to those trends observed in the lower vertebral levels (Table 2), with the exception of the dens, whose average BMD was approximately 200% larger than the other anatomical regions in the vertebra, see Fig 6. This was also in agreement with similar measurements conducted on young adult cervical spines [18]. Consistent with all of the other vertebral levels, the posterior region of C1 was characterized by a larger BMD than the anterior portion, see Table 1. To date, only Anderst et al. have analyzed the mineral density distribution in C1, and these investigators found the anterior C1 region to possess the highest BMD [18]. It should be noted that in their study, C1 was split in three anatomical regions which were, in order of BMD magnitudes, the anterior arch, the posterior arch and the lateral masses. In contrast, the anatomical partition utilized in this study combined the anterior arch and the lateral masses. This may explain the discrepancy between our results and those of Anderst et al. The BMD distribution of the vertebral bodies hereby reported agrees with computed tomographic osteoabsorptiometry measurements of mineral density in 80 cervical vertebral endplates which shows that density of the posterolateral region of the endplate was greater than that in the anterior region [32].

Some limitations must be noted. This analysis was based on 23 cervical spines. A larger sample size would allow further generalizing the results hereby reported. For instance, a larger

number of specimens would have allowed a multifactorial analysis (e.g., including vertebral level, gender, BMI, anatomical location, etc.) to identify those factors that are more influential on the distribution of BMD in cervical vertebrae. Nevertheless, our findings provide important preliminary insights on the BMD distribution in geriatric cervical spines and how this differentiates from that of young and adult spines. Furthermore, this study is limited by the chosen resolution of the CT and by the user discretion in identifying the landmarks. Both limitations are common to other similar densitometric studies that have been performed on clinically obtained CT data previously published [33,34]. The significant variations identified in this study will allow the development of deep learning algorithms targeted to the identification of the relevant volumes as performed in more recent studies [35].

In conclusion, the results of this study may suggest that gender could have an effect on both bone volume and density across all the levels of the cervical spine, with female having smaller vertebrae with higher BMD. There is a general agreement of the results of this contribution with those of previous studies. However, some age-related effects have also been observed: 1) the BMD of our elderly vertebrae is generally lower than that found in young and adult cohorts; 2) the BMD distribution across cervical levels in elderly is different from that of younger population. Finally, lateral-posterior regions of the vertebrae, including transverse processes, lateral masses, and spinous process regions for C3-C5, as well as in the dens for C2, were characterized by the highest values of BMD. Importantly, at each level, the posterior portion of the vertebral body possessed higher BMD that the anterior one. This information suggests that, in the elderly, surgical fixation of the posterior elements should be preferred to anterior ones.

## Supporting information

**S1 File.**
(XLSX)

## Author Contributions

**Conceptualization:** Giovanni F. Solitro, Kenrick C. Lam, Randal P. Morris, Abeer Albarghouthi, Ronald W. Lindsey, Loren L. Latta, Francesco Travascio.

**Data curation:** Ryan S. Garay, Francesco Travascio.

**Formal analysis:** Ryan S. Garay, Francesco Travascio.

**Investigation:** Abeer Albarghouthi, Ronald W. Lindsey, Loren L. Latta, Francesco Travascio.

**Methodology:** Giovanni F. Solitro, Kenrick C. Lam, Randal P. Morris, Ronald W. Lindsey, Loren L. Latta, Francesco Travascio.

**Software:** Ryan S. Garay.

**Writing – original draft:** Ryan S. Garay, Giovanni F. Solitro, Kenrick C. Lam, Randal P. Morris, Abeer Albarghouthi, Ronald W. Lindsey, Loren L. Latta, Francesco Travascio.

**Writing – review & editing:** Ryan S. Garay, Giovanni F. Solitro, Kenrick C. Lam, Randal P. Morris, Abeer Albarghouthi, Ronald W. Lindsey, Loren L. Latta, Francesco Travascio.

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
