## [Decision Letter · Decision Letter 0]

27 Apr 2022

PONE-D-22-03568Characterization of Regional Variation of Bone Mineral Density in the Geriatric Human Cervical Spine by Quantitative Computer TomographyPLOS ONE

Dear Dr. Travascio,

Thank you for submitting your manuscript to PLOS ONE. After careful consideration, we feel that it has merit but does not fully meet PLOS ONE’s publication criteria as it currently stands. Therefore, we invite you to submit a revised version of the manuscript that addresses each point raised during the review process. Both reviewers questioned statistical methods. Reviewer 1 pointed out inconsistencies in the abstract and study motivation, and suggested comparison to other methods and data in the literature. Reviewer 2 pointed out inconsistencies between results and conclusions.

We look forward to receiving your revised manuscript.

Kind regards,

Ryan K. Roeder, PhD

Academic Editor

PLOS ONE

Journal Requirements:

Reviewers' comments:

Reviewer's Responses to Questions

**Comments to the Author**

1. Is the manuscript technically sound, and do the data support the conclusions?

Reviewer #1: Yes

Reviewer #2: No

2. Has the statistical analysis been performed appropriately and rigorously? 

Reviewer #1: Yes

Reviewer #2: Yes

3. Have the authors made all data underlying the findings in their manuscript fully available?

Reviewer #1: Yes

Reviewer #2: Yes

4. Is the manuscript presented in an intelligible fashion and written in standard English?

Reviewer #1: Yes

Reviewer #2: Yes

5. Review Comments to the Author

Reviewer #1: The authors present a detailed compilation of cervical spine bone mineral density values apportioned by relevant anatomical sites in a geriatric cohort. Said apportioning is achieved through an elaborate workflow where the volumetric model obtained by segmentation is processed with a variety of CAD programs. The manuscript is well-written and its contribution of data for tissue sourced from older adult donors is welcome, as there is scarce data on this topic in the literature.

However, some questions arise after reading the manuscript:

The authors state that odontoid fractures as a motivator for this research, but do not include that region in the results. Were the specimens only consisting of the base of the cranium to C5 or did they include the whole head? The materials and methods section mentions orienting the cervical column in a supine position and that the qCT image volume “included the entire head”. Please clarify this for consistency’s sake. If the head was available, why not report the base of the cranium BMD as well? Lastly, it would have been more complete reporting if C6-C7 were included, but it seems that the specimens only spanned from C1 to C5.

The abstract mentions that BMD was reported in Hounsfield Units (which is a measurement of attenuation), but the tables show data converted to g/cm3 and the charts show data in mg/cm3. Make sure this is consistent across the document.

The zone subdivision (1 through 4) makes sense anatomically and clinically. Was there any other rationale to subdivide yet even further each vertebra into the 12 or 14 smaller sub-zones?

Data about the cohort that would be useful to have in a manuscript of this type are cause of death, BMI, bone quality (osteoporotic or osteopenic), and if possible, the cervical Cobb angles to characterize the spinal curvature. This last parameter would have helped to discuss the fact that C4 and C5 higher values were reported, presumably due to load.

While the method to obtain the data is scientifically sound, comparisons to other methodologies used for this purpose would be useful, especially the use of DXA which is the clinical gold-standard to evaluate bone quality. Another method that has not been discussed by the authors and is very accurate (which would also have avoided the elaborate virtual partitioning of the vertebrae) is CT Osteoabsorptiometry. In a seminal paper by Dr Muller-Gerbl (PMID 18193299), they report these very values for the cervical spine. Comparison against this method would also be useful.

The choice of aggregating data into groups ignoring gender or level is an interesting one. Was this due to the sample size? Was a power analysis conducted for this sample? Admittedly, this is not a large population study that can provide a very large sample size, but including these divisions by level, gender, BMI, anatomical location, etc. are useful when analyzing the data looking for which factor is the more influential.

In the Discussion section, the first paragraph mentions that head loads in both genders should be similar. That is the common assumption, however, body habitus and perhaps overall proportions may play a role in this. Do you have anthropomorphometric data to this effect? (not necessarily from your donors, that is probably difficult to trace), but in general?

In summary, this is a well-received contribution to the description of the cervical spine tissue material properties, but needs some minor modifications for completeness before the manuscript can be ready for publication.

Typos/Minor changes (please include line numbers next time, for easier reporting of edits)

The first sentence of the second paragraph in the Abstract should start with “Data trends suggest…”

Regression coefficient should be R^2 (superscript) instead of just "R2". (Journal review website does not allow superscript characters)

DISCUSSION subtitle currently reads DISUCSSION.

Figure captions, what are the error bars? (SD?)

Reviewer #2: This is a good study that expands the body of knowledge regarding BMD levels and distribution within the human spine, specifically the cervical vertebrae. The study is well-designed: the partitioning of the vertebrae helps to provide insight to BMD distribution in the analyzed vertebrae, and the statistical tests used are appropriate.

Revisions are suggested primarily due to some of the conclusions of the study. The final paragraph begins by stating that 'this study indicates that gender has an effect on bone volume and density across all levels of the cervical spine...' However, the study's statistical analysis did not find any significant differences in BMD values (p-value > 0.05). Given the relatively small sample size, and a p-value that is described only as greater than 0.05, this conclusion is not supported by the data. This conclusion also creates an inconsistency in the report, as the lack of difference between male and female BMD is used as justification for combining data samples at multiple points in the study.

In addition, the final paragraph claims that 'BMD distribution...in elderly is different from that of younger population, likely due to postural changes occurring with ageing.' I agree that it is certainly possible (even probable) that postural changes may affect this, but the way it is currently worded seems overstated, and that these changes are due solely to postural changes. Further studies/evidence would be needed to support the claim's current wording that these distribution changes are likely due to postural changes. Hormonal or metabolic changes associated with aging may play a significant or predominant role in this.

The main conclusion of the study seems to be that BMD is higher in the posterior regions of the vertebral body as compared to the anterior regions of the vertebral body. Mention could also be made that BMD was highest in the lateral/posterior regions of the vertebrae, including transverse processes/lateral masses and spinous process regions for C3-C5, as well as in the dens for C2.

6. PLOS authors have the option to publish the peer review history of their article (what does this mean?). If published, this will include your full peer review and any attached files.

Reviewer #1: No

Reviewer #2: **Yes: **Tyler C. Kreipke

---

## [Author Response · Author response to Decision Letter 0]

2 May 2022

Reviewers' comments:

We thank the Reviewers for the insightful comments and useful suggestions provided. We have conducted a revision of the manuscript as suggested. Changes throughout the text have been carried out as per Reviewers requests. In particular, the ‘Discussion’ section has been edited to include further discussion points and needed clarifications, as per Reviewers’ input. As a result, a new reference has been added to the manuscript. Also, the figures captions have been modified to specify the meaning of the bars. Minor changes throughout the manuscript have been implemented for consistency with the major changes included in this revision. The following are the detailed responses to the Reviewers’ comments. Revisions to the manuscript are highlighted in yellow.

Reviewer #1:

The authors present a detailed compilation of cervical spine bone mineral density values apportioned by relevant anatomical sites in a geriatric cohort. Said apportioning is achieved through an elaborate workflow where the volumetric model obtained by segmentation is processed with a variety of CAD programs. The manuscript is well-written and its contribution of data for tissue sourced from older adult donors is welcome, as there is scarce data on this topic in the literature. However, some questions arise after reading the manuscript: 

1. The authors state that odontoid fractures as a motivator for this research, but do not include that region in the results. Were the specimens only consisting of the base of the cranium to C5 or did they include the whole head? The materials and methods section mentions orienting the cervical column in a supine position and that the qCT image volume “included the entire head”. Please clarify this for consistency’s sake. If the head was available, why not report the base of the cranium BMD as well? 

Response: We would like to specify that with ‘odontoid’ we meant the ’odontoid process’ which is located in the anterior-superior portion of C2. We have clarified this in the abstract, as well as in the introduction. Our specimens also included the head, so also the cranium was scanned. While we agree on the value of documenting the bone mineral density at the base of the cranium, we did not report information on this anatomical region as our study focused on the cervical spine.

2. Lastly, it would have been more complete reporting if C6-C7 were included, but it seems that the specimens only spanned from C1 to C5.

Response: We completely agree. However, our samples did not include C6 and C7. This is because we focused on vertebral levels closer to C2, given that the most prevalent vertebral fractures in elderly occur at the odontoid process in C2. 

3. The abstract mentions that BMD was reported in Hounsfield Units (which is a measurement of attenuation), but the tables show data converted to g/cm3 and the charts show data in mg/cm3. Make sure this is consistent across the document.

Response: we edited the abstract to clarify that the BMD was calculated from the Hounsfield units via calibration phantom. This device allows us to convert HU into mg/cm3. See the Methods and Findings portion of the abstract, it reads: “The BMD was calculated from the Hounsfield units via calibration phantom”.

4. The zone subdivision (1 through 4) makes sense anatomically and clinically. Was there any other rationale to subdivide yet even further each vertebra into the 12 or 14 smaller sub-zones?

Response: C3-5 were split in 12 zones to document the BMD of the vertebral body (8 zones) and the lateral masses (2 zones) since these are the anatomical locations where hardware is generally implanted. Finally, BMD quality in posterior processes (2 more zones) is relevant to evaluate the potential quality of bone grafts. The C2 included 2 additional zones to specifically document the dens, which is a unique characteristic of this vertebra. We have added this clarification in the 3rd paragraph of ‘Discussion’, it reads: “The choice of the specific anatomical region subdivision was motivated to document BMD quality in the anterior and posterior vertebral body (8 regions), as well as the lateral masses (2 regions) since these are the locations where fixation hardware is usually implanted. Two additional regions (including the lamina and the spinous process) were also investigated as this is where bone grafts can be harvested from.”

5. Data about the cohort that would be useful to have in a manuscript of this type are cause of death, BMI, bone quality (osteoporotic or osteopenic), and if possible, the cervical Cobb angles to characterize the spinal curvature. This last parameter would have helped to discuss the fact that C4 and C5 higher values were reported, presumably due to load.

Response: The BMI values were included in the specimens description (see ‘Specimens’ subsection of Materials and Methods). Unfortunately, information on cause of death was not available. Bone quality data can be directly available from the measurements performed in this study, but classification in normal, osteopoenic and osteoporotic is not possible as these categories are not defined for cervical spine. Finally, it was not possible to determine the Cobb angle as our samples came already dissected from C5 level to the head. While we agree that this could have been a valuable information, we believe that the correlation of Cobb angle to BMD distribution would have been out of the scope of this work.

6. While the method to obtain the data is scientifically sound, comparisons to other methodologies used for this purpose would be useful, especially the use of DXA which is the clinical gold-standard to evaluate bone quality. Another method that has not been discussed by the authors and is very accurate (which would also have avoided the elaborate virtual partitioning of the vertebrae) is CT Osteoabsorptiometry. In a seminal paper by Dr Muller-Gerbl (PMID 18193299), they report these very values for the cervical spine. Comparison against this method would also be useful.

Response: We appreciate the suggestions of the Reviewer for improving the quality of this contribution. Unfortunately, DXA is not a routine evaluation tool in the cervical spine [see Yoganandan et al.2006, Bone]. Due to the anatomy of the lower cervical spine, measurements of the entire cervical spine are technically challenging with DXA due to projection artifacts [Korovessis et al.(1994) Eur Spine J; Ordway et al.(2007) Eur Spine J]. Therefore, to our best knowledge, there are no studies reporting DXA values of cervical vertebrae in humans. We also thank the Reviewer for bringing to our attention the important contribution of Muller-Gerbl and co-workers. We enriched our discussion by comparing our results of BMD distribution in the vertebral bodies to the CT osteoabsorptiometry measurements of bone mineral distribution in endplates measured by Muller-Gerbl et al., see the end of the 3rd paragraph of ‘Discussion’, it reads: “The BMD distribution of the vertebral bodies hereby reported agrees with computed tomographic osteoabsorptiometry measurements of mineral density in 80 cervical vertebral endplates which shows that density of the posterolateral region of the endplate was greater than that in the anterior region [35]”.

7. The choice of aggregating data into groups ignoring gender or level is an interesting one. Was this due to the sample size? Was a power analysis conducted for this sample? Admittedly, this is not a large population study that can provide a very large sample size, but including these divisions by level, gender, BMI, anatomical location, etc. are useful when analyzing the data looking for which factor is the more influential.

Response: All the data reported in the figures 5-7 are segregated by gender (male/female) and vertebral level (c1 to c5), and vertebral region. Since no statistically significant difference were found across gender and level (for the case of C3-C5), we decided to pool the measurements together to increase the sample size. Hence, with a convenience sample of 23 spines, we conducted a post hoc power analysis. We found that our power is larger than 90% when trying to identify differences in BMD across regions of the vertebral bodies, for all the levels investigated. As noted in the limitations of the analysis (see fourth paragraph of ‘Discussion’), the results provided in this study represent preliminary insights on the BMD distribution in geriatric cervical spines. Given the limited number of spines investigated, fragmentation of the data in additional subgroups including also BMI would have further reduced the sample size (and the power) for the statistical considerations reported. We have noted this limitation in the 4th paragraph of ‘Discussion’, it reads: “A larger sample size would allow further generalizing the results hereby reported. For instance, a larger number of specimens would have allowed a multifactorial analysis (e.g., including vertebral level, gender, BMI, anatomical location, etc.) to identify those factors that are more influential on the distribution of BMD in cervical vertebrae.”

8. In the Discussion section, the first paragraph mentions that head loads in both genders should be similar. That is the common assumption, however, body habitus and perhaps overall proportions may play a role in this. Do you have anthropomorphometric data to this effect? (not necessarily from your donors, that is probably difficult to trace), but in general?

Response: To our best knowledge, we do not have information on potential differences in cervical spine loading across genders. The statement on similarity of the loading magnitudes is a speculation proposed by Anderst and co-workers we referred to in our contribution. We agree with the Reviewer that other factors, like posture and habitus, may influence the loading of the cervical spine. We mitigated the statement in the discussion clarifying that “As speculated by Anderst and co-workers […], mechanical loads on female and male cervical spines might be similar”.

9. In summary, this is a well-received contribution to the description of the cervical spine tissue material properties, but needs some minor modifications for completeness before the manuscript can be ready for publication.

Response: We appreciate the comments of the Reviewer and believe that this revision process has significantly improved the quality of our work.

10. Typos/Minor changes (please include line numbers next time, for easier reporting of edits)

The first sentence of the second paragraph in the Abstract should start with “Data trends suggest…”

Response: done as suggested.

11. Regression coefficient should be R^2 (superscript) instead of just "R2". (Journal review website does not allow superscript characters)

Response: correction made. Thanks

12. DISCUSSION subtitle currently reads DISUCSSION.

Response: thank you for noticing that. Typo corrected.

13. Figure captions, what are the error bars? (SD?)

Response: figure captions for figures 3, 4, 5, 6 and 7 were modified to explain that the bar represents 1 standard deviation.

Reviewer #2

This is a good study that expands the body of knowledge regarding BMD levels and distribution within the human spine, specifically the cervical vertebrae. The study is well-designed: the partitioning of the vertebrae helps to provide insight to BMD distribution in the analyzed vertebrae, and the statistical tests used are appropriate. 

1. Revisions are suggested primarily due to some of the conclusions of the study. The final paragraph begins by stating that 'this study indicates that gender has an effect on bone volume and density across all levels of the cervical spine...' However, the study's statistical analysis did not find any significant differences in BMD values (p-value > 0.05). Given the relatively small sample size, and a p-value that is described only as greater than 0.05, this conclusion is not supported by the data. This conclusion also creates an inconsistency in the report, as the lack of difference between male and female BMD is used as justification for combining data samples at multiple points in the study.

Response: We agree with the Reviewer and edited the paragraph to better reflect the actual findings of this study. It reads: ”…the results of this study may suggest that gender could have an effect on both bone volume and density across all the levels of the cervical spine…”. 

2. In addition, the final paragraph claims that 'BMD distribution...in elderly is different from that of younger population, likely due to postural changes occurring with ageing.' I agree that it is certainly possible (even probable) that postural changes may affect this, but the way it is currently worded seems overstated, and that these changes are due solely to postural changes. Further studies/evidence would be needed to support the claim's current wording that these distribution changes are likely due to postural changes. Hormonal or metabolic changes associated with aging may play a significant or predominant role in this.

Response: We agree with the Reviewer and removed the statement from the manuscript. 

3. The main conclusion of the study seems to be that BMD is higher in the posterior regions of the vertebral body as compared to the anterior regions of the vertebral body. Mention could also be made that BMD was highest in the lateral/posterior regions of the vertebrae, including transverse processes/lateral masses and spinous process regions for C3-C5, as well as in the dens for C2.

Response: In agreement with the reviewer, we edited the discussion to integrate this observation, now it reads: “Finally, lateral-posterior regions of the vertebrae, including transverse processes, lateral masses, and spinous process regions for C3-C5, as well as in the dens for C2, were characterized by the highest values of BMD. Importantly, at each level, the posterior portion of the vertebral body possessed higher BMD that the anterior one. This information suggests that, in the elderly, surgical fixation of the posterior elements should be preferred to anterior ones.”

---

## [Decision Letter · Decision Letter 1]

27 Jun 2022

Characterization of Regional Variation of Bone Mineral Density in the Geriatric Human Cervical Spine by Quantitative Computer Tomography

PONE-D-22-03568R1

Dear Dr. Travascio,

We’re pleased to inform you that your manuscript has been judged scientifically suitable for publication and will be formally accepted for publication once it meets all outstanding technical requirements.

Kind regards,

James Mockridge

Staff Editor

PLOS ONE

Editor's comments:

Reviewer #2 has indicated that there are some minor text errors to correct, so please do these before submitting your final files. 

Reviewers' comments:

Reviewer's Responses to Questions

**Comments to the Author**

1. If the authors have adequately addressed your comments raised in a previous round of review and you feel that this manuscript is now acceptable for publication, you may indicate that here to bypass the “Comments to the Author” section, enter your conflict of interest statement in the “Confidential to Editor” section, and submit your "Accept" recommendation.

Reviewer #1: All comments have been addressed

Reviewer #2: All comments have been addressed

2. Is the manuscript technically sound, and do the data support the conclusions?

Reviewer #1: Yes

Reviewer #2: Yes

3. Has the statistical analysis been performed appropriately and rigorously? 

Reviewer #1: Yes

Reviewer #2: Yes

4. Have the authors made all data underlying the findings in their manuscript fully available?

Reviewer #1: Yes

Reviewer #2: Yes

5. Is the manuscript presented in an intelligible fashion and written in standard English?

Reviewer #1: Yes

Reviewer #2: Yes

6. Review Comments to the Author

Reviewer #1: The authors have addressed satisfactorily all comments. Thank you.

Reviewer #2: Comments and questions regarding the study have been satisfactorily addressed, and I believe that the authors have produced a quality study that will advance the knowledge of the field.

Upon reading the revised manuscript, I noted few small typos that could be addressed:

1. The title currently reads "...Quantitative Computer Tomography" instead of "...Quantitative Computed Tomography".

2. In the QCT Image Acquisition subsection of Materials and Methods, it reads "...0.5 x 0.5 mm in-plane pixel resolution..." instead of "...0.5 x 0.5 mm^2 in-plane pixel resolution...".

3. Near the end of the 2nd paragraph in Discussion, it reads "Wolff Law" instead of "Wolff's Law".

4. In the caption for Figure 3, there is no space between "male" and "(black)".

7. PLOS authors have the option to publish the peer review history of their article (what does this mean?). If published, this will include your full peer review and any attached files.

Reviewer #1: No

Reviewer #2: **Yes: **Tyler Kreipke

---

## [Editor Report · Acceptance letter]

29 Jun 2022

PONE-D-22-03568R1 

Characterization of Regional Variation of Bone Mineral Density in the Geriatric Human Cervical Spine by Quantitative Computer Tomography 

Dear Dr. Travascio:

I'm pleased to inform you that your manuscript has been deemed suitable for publication in PLOS ONE. Congratulations! Your manuscript is now with our production department. 

Kind regards, 

on behalf of

Dr James Mockridge 

Staff Editor

PLOS ONE